# Forgery Cyber-Attack Supported by LSTM Neural Network: An Experimental Case Study

**DOI:** 10.3390/s23156778

**Published:** 2023-07-28

**Authors:** Krzysztof Zarzycki, Patryk Chaber, Krzysztof Cabaj, Maciej Ławryńczuk, Piotr Marusak, Robert Nebeluk, Sebastian Plamowski, Andrzej Wojtulewicz

**Affiliations:** 1Institute of Control and Computation Engineering, Faculty of Electronics and Information Technology, Warsaw University of Technology, 00-665 Warsaw, Poland; krzysztof.zarzycki@pw.edu.pl (K.Z.); maciej.lawrynczuk@pw.edu.pl (M.Ł.); piotr.marusak@pw.edu.pl (P.M.); robert.nebeluk@pw.edu.pl (R.N.); sebastian.plamowski@pw.edu.pl (S.P.); andrzej.wojtulewicz@pw.edu.pl (A.W.); 2Institute of Computer Science, Faculty of Electronics and Information Technology, Warsaw University of Technology, 00-665 Warsaw, Poland; krzysztof.cabaj@pw.edu.pl

**Keywords:** cyber-security, cyber-attacks, LSTM neural networks, industrial control systems, SCADA, PLC

## Abstract

This work is concerned with the vulnerability of a network industrial control system to cyber-attacks, which is a critical issue nowadays. This is because an attack on a controlled process can damage or destroy it. These attacks use long short-term memory (LSTM) neural networks, which model dynamical processes. This means that the attacker may not know the physical nature of the process; an LSTM network is sufficient to mislead the process operator. Our experimental studies were conducted in an industrial control network containing a magnetic levitation process. The model training, evaluation, and structure selection are described. The chosen LSTM network very well mimicked the considered process. Finally, based on the obtained results, we formulated possible protection methods against the considered types of cyber-attack.

## 1. Introduction

Cyber-attacks pose an increasingly significant threat to industrial networks. This applies to industries such as energy, robotics, and automotive [1,2,3,4,5,6,7,8]. On the one hand, relatively obvious protection methods against such attacks exist, i.e., isolation of information technology (IT) and operational technology (OT) networks, as well as the precise definition of permissions for individual human and hardware elements of the network based on whitelisting [9]. On the other hand, unauthorized access to such networks’ critical elements remains a real threat [10].

Cyber-security and proper operation of control algorithms heavily depend on sensors required to provide precise measurements. First, when a cyberattack occurs, measurements from those sensors might be used to detect this malicious activity and further investigate the problem. There are various approaches to performing this detection, e.g., models such as support vector data description [11], neural autoencoders [12], long short-term memory networks [13], k-nearest-neighbors (KNN), decision tree (DT), support vector machines (SVM), naive Bayes and random forest [14] could be trained on the data collected from sensors during the typical work of the system. Then, when the measurements differ from the norm, the said algorithms can raise the alarm about anomalies or potential attacks. Anomalies are often related to some errors in the sensors, external disturbances, or even manual human interventions, but they can also arise from the interference of an malicious actor. The algorithms used for anomaly detection vary in complexity, from simple threshold analysis [15] to deep neural network models [16,17] or classifiers based on statistical properties [18,19,20].

Another category of cyber-attacks is attacks on the communication between sensors and other devices. This ranges from denial of service attacks [21], sleep deprivation attacks [22], through flooding [23], and jamming attacks [24], to attacks based on network traffic analysis and self-replay attacks [25]. All of these rely on the fact that the data from the sensors must be transmitted somehow. This is an especially significant threat to IoT and industrial IoT devices and sensors [26,27]. Even the commonly used IIoT protocol MQTT is prone to typical attacks [28,29], e.g., as considered in this paper, man in the middle attacks [30].

Second, the efficiency of control algorithms relies on the accuracy of sensors, e.g., in UAV [31] or autonomous vehicle [32,33] control. It is necessary to point out that the efficient operation of control algorithms can be significantly improved using neural networks. Typically, neural networks serve as models of dynamical processes, e.g., utilized online in model predictive control (MPC) algorithms [34,35].

The intensive development of neural networks in various configurations makes it possible to find networks that approximate dynamical phenomena [36,37]. Additionally, neural networks can carry out effective cyber-attacks that are invisible to the process operator with much less effort than classical methods. Therefore, this publication proposes defense mechanisms against such attacks that would provide additional protection against the destructive actions of the attacker in the event of a breach of the OT network’s integrity.

Often in research, it is assumed that the attacker has gained access to the industrial network, and further considerations are made from that point. This is because gaining access to the attacked network is often the result of social engineering, which is difficult to standardize. Specifying a precise and effective methodology to access an industrial network is also difficult. Therefore, assuming access has been obtained, the attacker may consider various methods of attacking the industrial network.

The operator often oversees the key time series or some derivative plots of this signal (e.g., histograms, signal increments, control errors). This paper presents the vulnerability directly related to the operator’s responsibilities. The attack presented in our work aims to take advantage of the operator’s inability to discern small changes in the signal, as the LSTM model that we propose generates similar results to the real signals. On the other hand, a slow degradation of the control quality is performed via subtle changes in the control signals that affect the signal that is being forged.

A commonly considered approach for detecting attacks based on network traffic is to use genetic or evolutionary algorithms [38], Bayesian networks [39], or machine learning [40]. One of the more time-consuming approaches is to identify the target under attack by monitoring network traffic and, based on that traffic or the data acquired from it, to develop a model of the process under attack, to accurately generate malicious signals. Such an approach is often impossible to implement, due to the large number of signals used in the network. In this case, it is necessary to identify signals relevant to the operation of the process and interpret them correctly. The model can be obtained using classical approaches such as support vector machines [10] or random forest [39]. A relatively new approach is to use deep neural networks [6,39]. These are attractive because of their high degree of automation of the learning process and because they require less knowledge of the process being modeled. However, these models tend to be less well adapted to the actual process, and their learning requires selecting a number of meta-parameters.

This study presents an example of using LSTM-type neural networks to model a process of unknown physical form. This model is then used to simulate the process. Such data are transmitted to the supervisory control and data acquisition (SCADA) system, to conceal the attack on the actual process. Finally, considerations are formulated regarding potential protection methods against such attacks, abstracting from the methodology for acquiring the process model.

### 1.1. Problem Definition: Cyber-Attacks

The problem considered in this paper relates to cyber-attacks, which are understood as harmful actions against critical elements of industrial processes accessible from the industrial network. They may lead to a prolonged reduction in the efficiency of industrial processes [41] or their accelerated aging [42]. Attacks aimed at immediately disabling network elements [43,44] are not considered in this context, as such a failure is easy to detect and repair. At the same time, the attacker may seek to remain hidden in their actions for as long as possible [45]. In these studies, the assumed motive of the attacker is to carry out a prolonged attack that would be difficult to detect by operators.

There are a few potential implications of carrying out prolonged cyber-attacks that are difficult for operators to detect:introduction of an ill-natured (e.g., high frequency) control signal, which might increase the aging rate of the actuators, which would result in a larger overall loss due to the prolonged exposure, compared to more devastating but quicker to detect and fix attacks, e.g., denial of service;gaining knowledge about the process of control; as the operators are not aware of the subtle changes in the control signals, subtle responses visible in the measurements might be attributed to noise and disturbances, whereas the attacker might use this knowledge to understand the process and design more precise attacks;the prolonged presence of the attacker in the network might result in getting operators used to the new characteristics of the signals, thus lowering the operators’ caution.

This paper considers an attack on one of the control loops in a system. It is worth underlining that the described system is simple, and thus there is a clear relation between the signals. From the point of view of the malicious actor, the control signal should be considered the most relevant for the attack, as the quality of this signal is often not validated. This is because this signal is a result of simple computations (e.g., a PID algorithm) based on the measured and setpoint values, and therefore it is assumed that the root cause is based on the measured or setpoint signals. There is, of course, a challenge for the attacker in determining which of the signals is the control signal (as designers or even maintainers of the system, this information should be known). There might be some premises for some signals to be assumed to be the control signals (e.g., dynamics of the signal, the variance of the signal increments). Nevertheless, in systems with hierarchical or generally complex control schemes, this might be nontrivial or impossible to achieve. In this paper, the interpretation of the signals is assumed to be known.

### 1.2. State of the Art: LSTM Neural Networks

Long short-term memory (LSTM) networks originated as a modification of the classical recurrent neural network (RNN), differing from it in the neuron’s structure [46]. Classical RNNs are prone to a vanishing gradient phenomenon [47], which often makes them unable to model long-term dependencies in data. LSTM networks, on the other hand, are much more resistant to this problem. The neuron in an LSTM network is often referred to as a cell and consists of a set of gates that regulate the flow of information. In particular, gates allow for selecting relevant, and forgetting unnecessary, information. This often occurs in classical recurrent networks. The unique memory properties of LSTM models have led them to find a wide range of applications, such as in traffic forecasting [48], gaze-based deception detection [49], speech synthesis [50], and handwriting recognition [51]. LSTMs can also be used to model nonlinear dynamic processes [52]. In this case, one can find that LSTM models with a relatively low number of neurons and internal weights, and short input sequences, are easily trained and offer excellent modeling quality. LSTM models can also be successfully implemented in a model predictive control algorithm [35,53,54]. LSTM networks are often used for time series modeling, i.e., for prediction of indoor temperature and air pollution [55], forecasting in finances [56], and time series classification [57].

### 1.3. Article Contribution

This work investigates the vulnerability of a physical industrial control system operating in a network to cyber-attacks. Specifically, we consider to what extent a cyber-attack can cause disruptions to the process operation but remain invisible to the operator. The Stuxnet worm performed the first such hidden attack, infecting PLC devices controlling Iran’s nuclear centrifuges. In effect, a hostile PLC program deliberately changed the spinning frequency, to destroy the centrifuge. However, normal signals were provided to the SCADA and the operator. In effect, hostile activity was detected when the centrifuge was seriously damaged. During our research, we would like to use machine learning methods to carry out such cyber attacks, while generating fake signals. A cyber attack that is difficult to detect while causing damage to a process can be challenging to carry out using classical methods, without knowledge of the system under attack. However, the development of artificial intelligence carries the risk of using machine learning methods to automate the execution of an attack. In this article, we examine whether LSTM networks allow a physical process to be modeled in an automated manner. Then, we use the process knowledge in this black-box model to mask the cyber attack. Security measures are also discussed, including data redundancy and isolating the industrial network from the IT network; the described attack is only possible in the case of a human error.

### 1.4. Article Organization

This work is structured as follows. First, Section 2 describes the laboratory control network used in our study. Section 3 presents the article’s main contribution: the LSTM neural networks for SCADA system operator deception; issues such as the network architecture, data-driven training, evaluation, and structure selection are discussed. Section 4 details the experimental evaluation of the obtained LSTM neural networks in the SCADA system and formulates possible protection mechanisms against cyber-attacks. Finally, Section 5 concludes the article.

## 2. Laboratory Process Control Network

To perform tests using algorithms considered in this paper, a laboratory process control network was designed and implemented. This is a controlled part of industrial reality, mainly focused on the industrial network’s OT and control network layers. This network environment was prepared with a few assumptions in mind. First, it should allow testing of a range of standard network protocols in the industry. Second, it should consist of various processes (fast and slow, simple and complex, stable and unstable, binary and continuous). Third, it should be implemented with the best industrial practices in mind.

Figure 1 depicts the laboratory process control network, which consists of several control processes, namely: a heating-cooling test stand (slow, simple, stable, continuous process) [58], MPS Festo stand (fast, complex, stable, binary process) [59,60,61], and a magnetic levitation (MAGLEV) process using INTECO (quick, simple, unstable, continuous process) [62,63]. Out of those, only the magnetic levitation process is considered in this paper. The following programmable logic controllers (PLCs) manage these processes: Mitsubishi Electric FX5, Mitsubishi Electric iQ-R, and Siemens S7-1200. In addition, each PLC was connected to a corresponding human–machine interface (HMI) display. All these network devices were connected to a simple switch that directed communication from a specific part of the network to the main managed switch. This network includes two SCADA systems: SCADA WinCC RT and SCADA MAPS; which were installed and run on separate PCs.

This paper considers a section of the whole workspace, namely the MAGLEV section, including the main managed switch, which connects all the simple switches in the network. In Figure 2, a photo shows the MAGLEV and heating–cooling sections of the laboratory workspace. The Mitsubishi iQ-R Series PLC on the left side of the photo is connected via an Inteco Power Unit to the MAGLEV, visible on the right side. The overall data flow of this part of the network is depicted in Figure 3. The HMI displays were not crucial in terms of this research, yet they are included for the sake of completeness. The control algorithm for the MAGLEV system consists of a PID controller with a precisely calibrated function for transforming raw measurements (voltage based on the amount of light received by the light sensor) into the actual distance of the levitating ball from the electromagnet.

The main communication route is between iQ-R and SCADA MAPS. The communication between the iQ-R and the MAGLEV process is based on pulse width modulation (PWM), to control the top coils’ power and used to pull the metal ball towards the coil, and analog signals, to measure the ball’s position. Therefore, this cannot be utilized to perform an attack. On the other hand, the communication with SCADA goes through a main managed switch and therefore is a good candidate for such an attack. The data sent between those two units only consist of requests for values of iQ-R registers. The SCADA system requests a few variables: the ball position, set-point value (calculated by the PLC), and the width of the PWM, i.e., common control signals. No additional information about the controller parameters was considered in this scenario.

The part of the laboratory workspace considered here allows testing the vulnerability of selected communication protocols. Here, we focused on the SLMP protocol, which allows communication between PLCs and the SCADA MAPS system.

It should be noted that, even though we consider SCADA MAPS in this workspace the SCADA of choice, in the end, we created an implementation of the SCADA system because of the inaccuracies of the SCADA MAPS. Mainly, the communication period was inconsistent, which caused confusing errors during the validation stages of the models. Next, despite being responded to, many messages were ignored and displayed as nonexistent, thus creating an even more confusing picture of the experiment results. The main job of the SCADA finally used in this paper was to send periodic requests for the values of some registers and store their values in files. Those files could be further displayed.

## 3. LSTM Neural Networks for SCADA System Operator Deception

### 3.1. Architecture of the LSTM Neural Network

The considered LSTM network has one hidden layer of $n_{N}$ neurons. Let *u* and *y* stand for the process input and output signals, respectively, while *k* denotes the discrete time, i.e., k=0,1,2,… For time series prediction, the input vector of the network is
(1)$$x\left(k\right)={\left[y\left(k-1\right)\;\cdots \;y\left(k-n_{A}\right)\right]}^{T}$$
where $n_{A}$ is an integer number. Alternatively, when the process’s set point is considered, the input vector is
(2)$$x\left(k\right)={\left[y\left(k-1\right)\;\cdots \;y\left(k-n_{A}\right)\;y^{\mathrm{SP}}\left(k\right)\right]}^{T}$$

Each LSTM neuron, often called a cell, consists of four gates: forget gate *f*, cell candidate gate *g*, input gate *i*, and output gate *o*. When the whole layer of LSTM containing $n_{N}$ cells is discussed, the gates can be represented as vectors f, g, i, o, each of dimensionality $n_{N}\times 1$. Weights related to the network input signals and past hidden state are denoted as W and R, respectively. The bias vector is denoted as b. Subscripts f, g, i, and o inform about the relation between the gates and weights. The weight matrices of the whole LSTM layer are
(3)$$W=\left[\begin{array}{c} W_{i}\\ W_{f}\\ W_{g}\\ W_{o} \end{array}\right],\;R=\left[\begin{array}{c} R_{i}\\ R_{f}\\ R_{g}\\ R_{o} \end{array}\right],\;b=\left[\begin{array}{c} b_{i}\\ b_{f}\\ b_{g}\\ b_{o} \end{array}\right]$$

At the time instant *k*, the LSTM layer first performs all gate computations
(4)$$\begin{array}{cc} i(k)= & \sigma (W_{i}x\left(k\right)+R_{i}h\left(k-1\right)+b_{i}) \end{array}$$
(5)$$\begin{array}{cc} f(k)= & \sigma (W_{f}x\left(k\right)+R_{f}h\left(k-1\right)+b_{f}) \end{array}$$
(6)$$\begin{array}{cc} g(k)= & \tanh(W_{g}x\left(k\right)+R_{g}h\left(k-1\right)+b_{g}) \end{array}$$
(7)$$\begin{array}{cc} o(k)= & \sigma (W_{o}x\left(k\right)+R_{o}h\left(k-1\right)+b_{o}) \end{array}$$

Next, the current cell state is calculated
(8)$$\begin{array}{c} c(k)=f(k)\circ c(k-1)+i(k)\circ g(k) \end{array}$$
where ∘ denotes the Hadamard product. Finally, the hidden state of the LSTM layer can be computed:(9)$$\begin{array}{c} h(k)=o(k)\circ \tanh(c(k)) \end{array}$$

The hidden state of the LSTM layer then enters a linear (fully-connected) layer. The linear layer has its own weight vector $W_{y}$ and scalar bias $b_{y}$. The output of the whole LSTM network at the time instant *k* can be computed as
(10)$$\begin{array}{c} y_{\mathrm{LSTM}}\left(k\right)=W_{y}h\left(k\right)+b_{y} \end{array}$$

Figure 4 shows the overall structure of the neural network used and the internal configuration of the LSTM cell.

### 3.2. Data-Driven Training, Evaluation, and Structure Selection of the LSTM Neural Network

The data set used for LSTM training and validation was obtained from the Laboratory Process Control Network. The data were collected using a custom-designed Proxy program, which was assumed to be placed between the PLC and SCADA software (Figure 5). The Proxy works in one of two modes: transparent or active. When in the transparent mode, the Proxy transmits messages from SCADA right to the PLC and all messages (including responses for SCADA requests) from PLC to SCADA. In this mode, the Proxy is used to listen and gather data that will later be used to train the LSTM model. The active mode hijacks the communication between the SCADA and PLC. The Proxy in this mode acts as a PLC and responds to the SCADA with forged signals, using a trained LSTM model, but it can also attack the PLC. This approach guarantees consistency between the data used for training and the environment in which the forgery attack is performed.

LSTM models were trained in MATLAB on a PC with Nvidia GeForce 970 GTX GPU, Intel i5-3450 CPU, and 16 GB of RAM. We used the Adam optimization algorithm, with a learning rate 0.001, a maximum number of 500 training epochs, and sequence padding-left direction enabled.

Training was performed for four categories of model:1.Models whose objective is to approximate the dynamics of the process when a sinusoidal set-point signal of the output is used; these models utilize previous values of the output signals to define the input vector of the LSTM network (Equation (Equation 1)), named hereafter ${\mathrm{LSTM}}_{\sin}$;2.Models whose objective is to approximate the dynamics of the process when a square set-point signal of the output is used; these models utilize previous values of the output signals to define the input vector of the LSTM network (Equation (Equation 1)), named hereafter ${\mathrm{LSTM}}_{\mathrm{sq}}$;3.Models whose objective is to approximate the dynamics of the process when a sinusoidal set-point signal of the output is used; these models utilize previous values of the output signals and the set-point signal to define the input vector of the LSTM network (Equation (Equation 2)), named hereafter ${\mathrm{LSTM}}_{\sin}^{\mathrm{SP}}$;4.Models whose objective is to approximate the dynamics of the process when a square set-point signal of the output is used; these models utilize previous values of the output signals and the set-point signal to define the input vector of the LSTM network (Equation (Equation 2)), named hereafter ${\mathrm{LSTM}}_{\mathrm{sq}}^{\mathrm{SP}}$;

The data set was collected from the laboratory stand. The data set was divided into training and validation data, each containing 8000 data samples. The average sampling time of the data equaled 100 ms. The procedure of training was as follows:1.First, all the data collected from the network were transformed from hexadecimal to decimal format and normalized. We transformed all the signals to be in a range from 0 to 1, except the set-point for the square signal, which was transformed to 1 and −1;2.Models with $n_{N}$ = 2 neurons and $n_{A}$ = 1 were trained. For each configuration of parameters, five models were trained;3.The order of dynamics was increased to $n_{A}$ = 8 and $n_{A}$ = 3, and 10 new models were trained;4.Steps 2 and 3 were repeated for models with $n_{N}$ equal to 12, 16, 24, 32, and 64.

All models were then tested on a test data set. The mean squared error (MSE) was used to measure the quality of the model.
(11)$$\mathrm{MSE}=\frac{1}{n}\sum_{k=1}^{n}{\left(y\left(k\right)-y_{\mathrm{LSTM}}\left(k\right)\right)}^{2}$$
where y(k) and $y_{\mathrm{LSTM}}\left(k\right)$ stand for the data sample and the model output for the sampling time *k*, respectively, and n=8000 is the number of samples.

Table 1 presents the error of the best performing LSTM model for each neuron number and the order of dynamics configuration for a sinusoidal ball movement. It can be seen that, for both ${\mathrm{LSTM}}_{\sin}$ and ${\mathrm{LSTM}}_{\sin}^{\mathrm{SP}}$ models, the error decreased as the number of neurons in the network increased. In both cases, the models with 64 hidden neurons performed best. A further increase in the number of neurons resulted in a more significant reduction in error; however, this also increased the computational cost of the model. Therefore, we decided that 64 neurons were sufficient for the case of a sinusoidal output signal. Let us discuss the effect of changing the order of model dynamics. For ${\mathrm{LSTM}}_{\sin}$ models, the error increased for higher orders of dynamics. However, for ${\mathrm{LSTM}}_{\sin}^{\mathrm{SP}}$ models, the best modeling quality was obtained for models with a dynamic order $n_{A}=3$. The ${\mathrm{LSTM}}_{\sin}^{\mathrm{SP}}$ models provided a better modeling quality compared to ${\mathrm{LSTM}}_{\sin}$ models. This is unsurprising, as LSTM models are presented with a more difficult task, having to operate fully in recursive mode. On the other hand, ${\mathrm{LSTM}}_{\sin}^{\mathrm{SP}}$ models take the set-point of the process as an input signal of the LSTM network, making it easier to match the frequency of the signal being modeled.

Table 2 shows the MSE of LSTM models for the square process output signal. For ${\mathrm{LSTM}}_{\mathrm{sq}}$ models, a better modeling quality was provided by those with more neurons. However, for ${\mathrm{LSTM}}_{\mathrm{sq}}^{\mathrm{SP}}$ models, one of the simplest models with eight neurons had the lowest error. The best models had a high order of dynamics, i.e., $n_{A}=3$. On the other hand, for ${\mathrm{LSTM}}_{\mathrm{sq}}^{\mathrm{SP}}$ models, there are often (e.g., for 12 or 24 neurons) cases where models with a lower order of dynamics are better. Again, it can be observed that adding the set-point as an input to the neural network significantly reduced the model errors.

A comparison of the performance of selected models (marked in blue in the tables) with the lowest error can be seen in Figure 6. In the case of ${\mathrm{LSTM}}_{\sin}$ models, the low MSE values are misleading, i.e., the model output shifted in phase with respect to the data after a short time. The ${\mathrm{LSTM}}_{\sin}^{\mathrm{SP}}$ model did not have this disadvantage. In its case, the model output was always well synchronized with the data. The outputs of the ${\mathrm{LSTM}}_{\mathrm{sq}}$ and ${\mathrm{LSTM}}_{\mathrm{sq}}^{\mathrm{SP}}$ models were very close to each other, which means that these models reproduced the data well.

## 4. Experimental Evaluation of LSTM Neural Networks in the SCADA System

The four models with the best quality were tested experimentally. A Python script read the data from the PLC and used LSTM models to generate output data, which were then displayed in SCADA.

The results of the experiments are shown in Figure 7. The output of the ${\mathrm{LSTM}}_{\sin}$ model reproduced the amplitude of the real signal well. However, it became out of sync with the process output relatively quickly. The ${\mathrm{LSTM}}_{\sin}^{\mathrm{SP}}$ model output was very well-fitted to the real data from the PLC. The outputs of the ${\mathrm{LSTM}}_{\mathrm{sq}}$ and ${\mathrm{LSTM}}_{\mathrm{sq}}^{\mathrm{SP}}$ models for the first 50 s of the models’ operation were very similar to each other and well-fitted to the real signal. However, Figure 8 shows the outputs of both models after 200 s of operation. Here, the advantage of the ${\mathrm{LSTM}}_{\mathrm{sq}}^{\mathrm{SP}}$ model is highlighted. This model was still in sync with the data after this time, while the ${\mathrm{LSTM}}_{\mathrm{sq}}$ model had slightly shifted in phase.

### 4.1. Cyber-Attack Description

The attack conducted during this experiment took advantage of the fact that most PLC devices do not offer reasonable authentication and authorization mechanisms. The most common security control is based on the configured number of concurrent sessions between the SCADA system or master PLC and the victim PLC. When at least one configured connection is not properly established, anybody with access to the industrial control system network can connect to the PLC and manipulate its registers. There are at least a few situations in which the PLC does not utilize all configured connections; for example, an incorrect configuration, change in the control process, temporal disconnection of engineering station, and hostile activity, which actively disconnects the legitimate connections. Regardless of the reason, all the described situations allow an attacker to control PLC registers.

Manipulation of PLC registers directly impacts the controlled MAGLEV process. During the experiments using custom tools, we manipulated the following chosen registers: D100, which contains the measured photodetector value corresponding to the ball position, and D1002, which contains the value of the width of PWM in the range of 0 (electromagnet constantly on) to 2000 (electromagnet constantly off). The custom tool connects to the PLC using SLMP (seamless message protocol) and overwrites the chosen register with the provided value as frequently as possible. Since these values are overwritten by the attacker more frequently than by the controller, the normal activity of the ball is disturbed. In the first case, the attack value 14,000 was written to the register D100. After a few seconds, the ball started “shaking” and stuck to the magnet in a few tens of seconds. In the second case, a value of 2000 was written to register D1002. After a while, the ball went out of the range of the electromagnetic field of the electromagnet and dropped.

### 4.2. Protection (Precautions) against Cyber-Attacks

To avoid the discussed method of attack, one might want to consider simple software protections. The main approach to prevent such attacks is introducing redundancy into the system. Plain duplication of the communication channel used to obtain certain information might not be enough. Thus, it is encouraged to use different protocols or at least introduce some easily revertible modifications of the original signal, e.g., using affine transformation or delaying the signal. This approach would require modifications on both sides of the communication, i.e., PLC and SCADA. One might also consider using communication protocols that implement encryption; this would prevent intruders from accessing the data.

It is possible to modify only the SCADA system to detect malicious actions such as those discussed in this paper. By constantly monitoring the control error, it is possible to detect controlled signals going out of phase (assuming a periodical set-point value), as is the case for models that do not use set-point values as inputs. It is also worth mentioning that noise and disturbance influence the controlled variable, which is often difficult to model without knowledge of the overall control process. Therefore, measuring the similarity of the consecutive periods of periodic signals might suggest the occurrence of a third party in the OT network. It would also be useful to implement anomaly detection algorithms or even calculate certain signals’ statistic properties (such as variance, skewness, or kurtosis) for the operator to observe, which would be useful for detecting unnatural behavior. As the operator might not be able to detect small changes in the behavior of the plant, these simple methods would amplify these kinds of observations for the operator.

Another useful approach would be forcing the SCADA system to periodically send information to the PLC about certain measurements of key signals (e.g., mean, variation in maximum and minimum values), which would then be calculated on the PLC side and tested for discrepancies. A significant difference in results implies the presence of an ill-intentioned entity in the communication.

There are other more obvious approaches to protecting an OT network. Utilizing secure protocols inside the OT network or isolating the network from the rest of the world are the most common ways to ensure secure data transfer. The challenge is to implement this in a convenient manner and that provides sufficient security to protect against hackers. The vulnerability of an OT network is often the result of human error. Therefore, security measures must not hinder employees’ workflow. Otherwise, they might unknowingly introduce holes in the security layer of the industrial network.

An interesting option is to use LSTM neural networks to establish a protection mechanism against cyber-attacks [64,65,66]. LSTMs can learn the temporal behavior of time series sequences. In the cited works, they helped to neutralize the corrupted measurements. As a result, the process received corrected signals, which increased the overall performance.

## 5. Conclusions

This work studied the vulnerability of a network industrial control system to cyber-attacks. The MAGLEV process was considered in our study. This process must be very fast, unstable, and nonlinear, with a short sampling time. We used LSTM neural networks to approximate the process dynamics. The model training, evaluation, and selection are described. The model simulated the process, and the output signal was transmitted to the SCADA system, to conceal the attack. We studied if, and to what extent, a cyber-attack could cause disruptions in the process operation. For the considered MAGLEV process, the LSTM networks enabled performing such cyber-attacks. Having completed many experiments, we formulated possible protection methods against the considered type of cyber-attacks.

It is necessary to emphasize that, in general, an LSTM neural network is a great universal tool, capable of approximating the behavior of dynamical systems. On the other hand, the training of LSTMs, network architecture selection, and network validation have to be performed for specific laboratory processes. We plan to compare the efficiency of different neural networks, e.g., gated recurrent units (GRUs).

It is worth underlining that the operator is often not an appropriate tool for data analysis. Specifically, the more subtle the changes are, the more likely the operator will not notice those changes. In addition, the SLMP protocol utilized in this paper is not sufficiently secure, allowing the attacker to easily modify the values of signals visible in the SCADA on the fly and use them with malicious intent. Both conclusions can be further extrapolated for other protocols, e.g., Modbus TCP, where the security concerns are not prioritized. This paper emphasizes the lack of security in the OT network, relying on the protection provided by network isolation from the outside world.

## Figures and Tables

**Figure 1 sensors-23-06778-f001:**
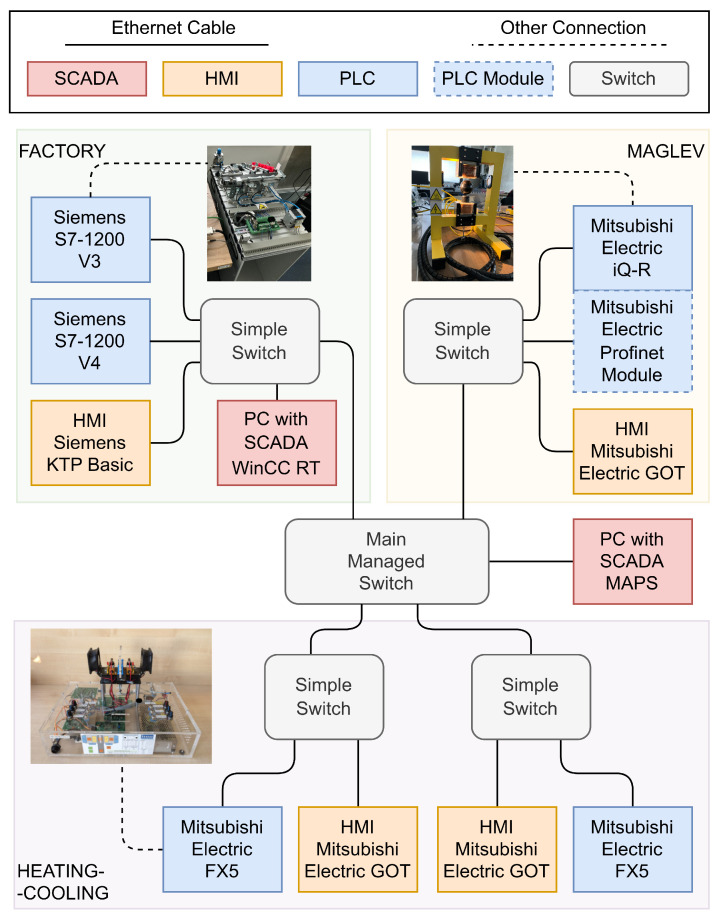
Schematic representation of connections within the laboratory control network.

**Figure 2 sensors-23-06778-f002:**
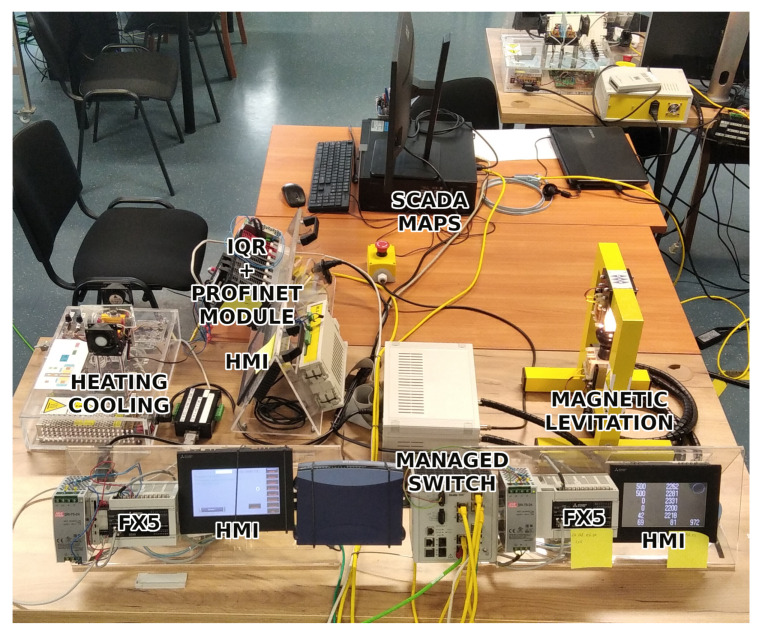
Photo showing MAGLEV and heating–cooling sections of the workspace, with the main managed switch.

**Figure 3 sensors-23-06778-f003:**
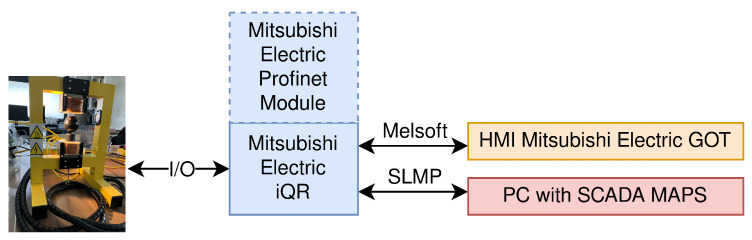
Schematic view of the data flow and protocols used in the considered part of the laboratory workspace.

**Figure 4 sensors-23-06778-f004:**
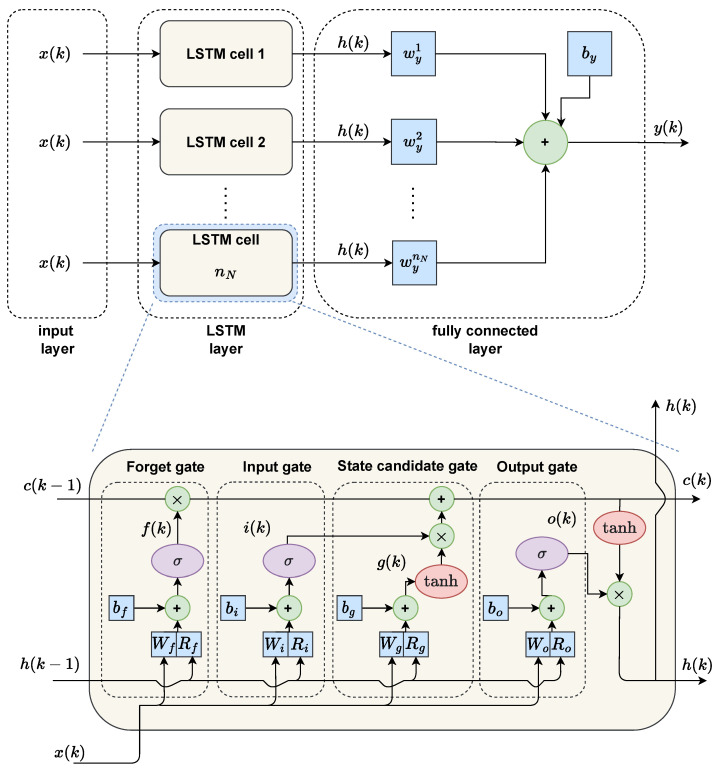
The overall structure of the neural network used (**top**), and the internal configuration of the LSTM cell (**bottom**). Blue indicates the model’s internal weights, green the algebraic operations, and purple and red the sigmoidal and hyperbolic tangent activation functions, respectively.

**Figure 5 sensors-23-06778-f005:**
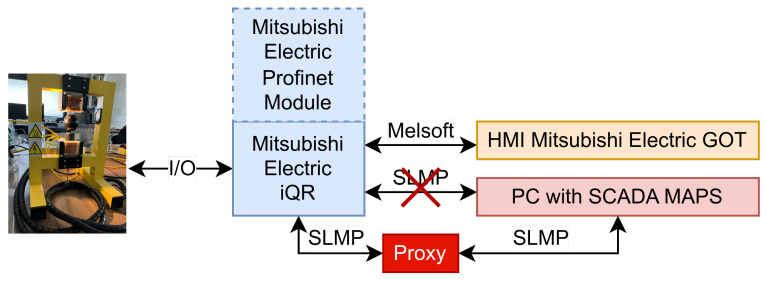
Schematic view of the data flow and protocols used in the considered part of the laboratory workspace when the Proxy has hijacked the communication.

**Figure 6 sensors-23-06778-f006:**
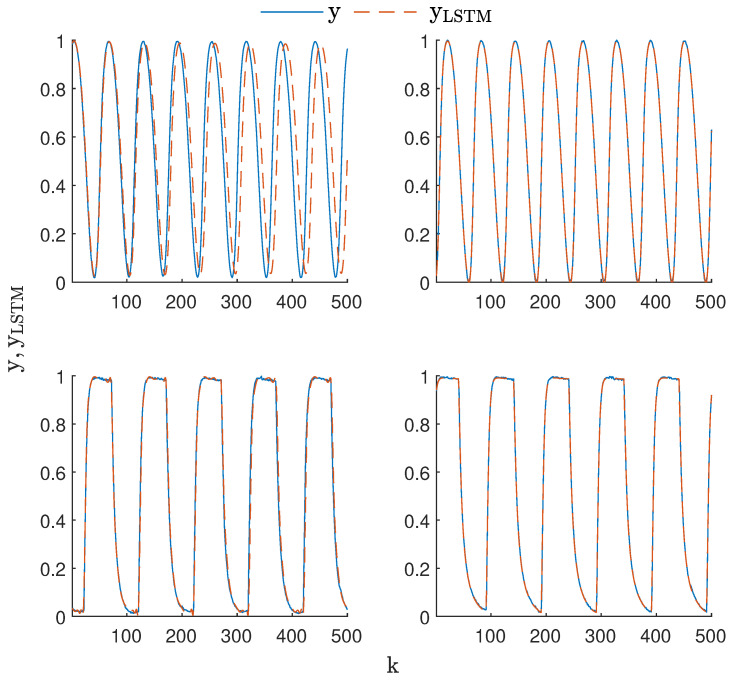
Best performing model outputs compared to the data from the validation data set: ${\mathrm{LSTM}}_{\sin}$ with $n_{N}=64$ and $n_{A}=1$ (**top-left**), ${\mathrm{LSTM}}_{\sin}^{\mathrm{SP}}$ with $n_{N}=64$ and $n_{A}=3$ (**top-right**), ${\mathrm{LSTM}}_{\mathrm{sq}}$ with $n_{N}=64$ and $n_{A}=3$ (**bottom-left**) and ${\mathrm{LSTM}}_{\mathrm{sq}}^{\mathrm{SP}}$ with $n_{N}=8$ and $n_{A}=3$ (**bottom-right**).

**Figure 7 sensors-23-06778-f007:**
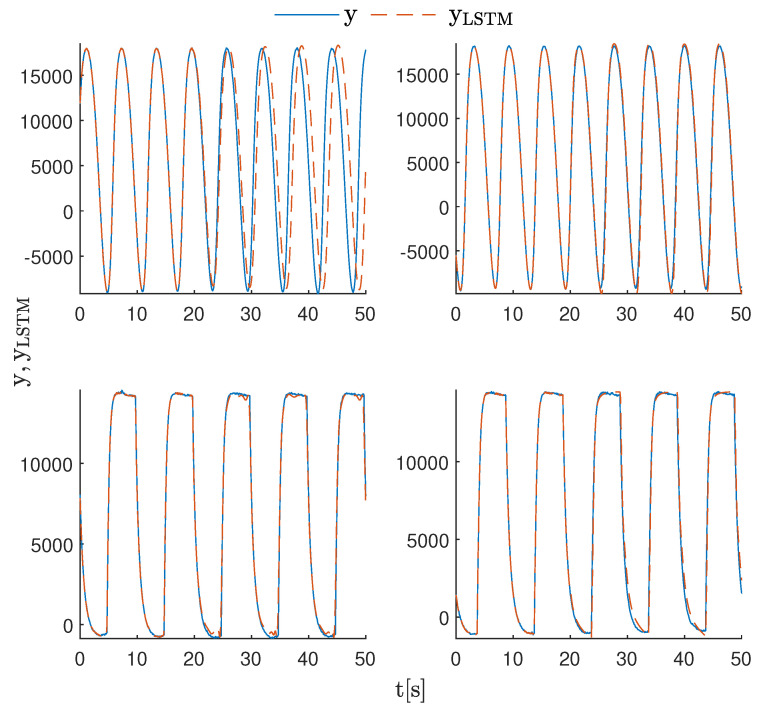
Best performing models working online, compared to real data collected from the PLC: ${\mathrm{LSTM}}_{\sin}$ with $n_{N}=64$ and $n_{A}=1$ (**top-left**), ${\mathrm{LSTM}}_{\sin}^{\mathrm{SP}}$ with $n_{N}=64$ and $n_{A}=3$ (**top-right**), ${\mathrm{LSTM}}_{\mathrm{sq}}$ with $n_{N}=64$ and $n_{A}=3$ (**bottom-left**) and ${\mathrm{LSTM}}_{\mathrm{sq}}^{\mathrm{SP}}$ with $n_{N}=8$ and $n_{A}=3$ (**bottom-right**).

**Figure 8 sensors-23-06778-f008:**
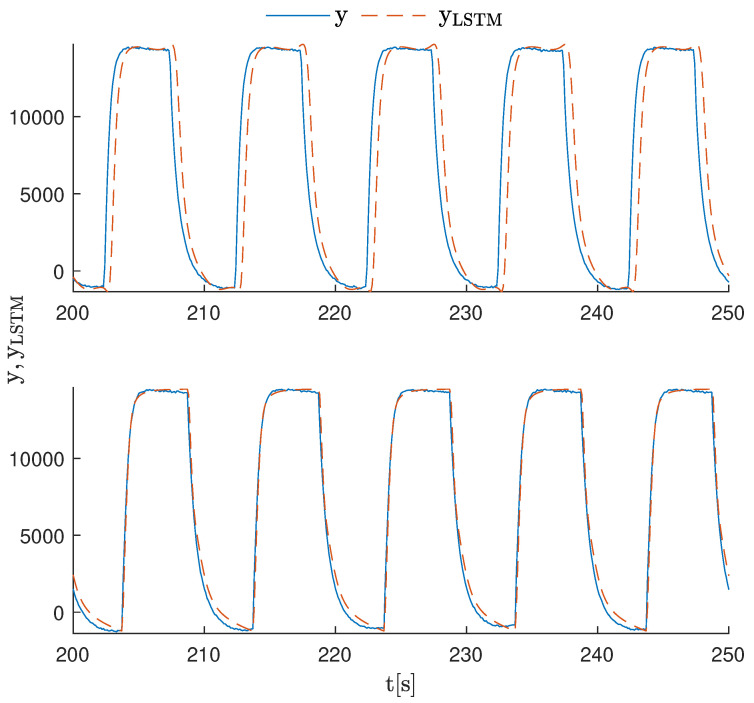
Comparison between the ${\mathrm{LSTM}}_{\mathrm{sq}}$ and ${\mathrm{LSTM}}_{\mathrm{sq}}^{\mathrm{SP}}$ models working online vs. real data collected from the PLC after 200 s of continuous operation.

**Table 1 sensors-23-06778-t001:** MSE errors for LSTM models for the sinusoidal set-point signal. The blue background indicates the best-performing models of each type with the lowest error value.

	${\mathrm{LSTM}}_{\sin}$				${\mathrm{LSTM}}_{\sin}^{\mathrm{SP}}$		
$n_{N}$	$n_{A}=1$	$n_{A}=2$	$n_{A}=3$		$n_{A}=1$	$n_{A}=2$	$n_{A}=3$
8	2.1 × 10 ^−1^	8.5 × 10 ^−3^	6.3 × 10 ^−4^		4.6 × 10 ^−4^	8.5 × 10 ^−4^	2.5 × 10 ^−4^
12	1.0 × 10 ^−3^	5.1 × 10 ^−4^	2.3 × 10 ^−1^		4.7 × 10 ^−4^	7.5 × 10 ^−4^	1.9 × 10 ^−4^
16	6.5 × 10 ^−4^	1.6 × 10 ^−3^	7.6 × 10 ^−4^		4.7 × 10 ^−4^	2.9 × 10 ^−4^	2.0 × 10 ^−4^
24	1.1 × 10 ^−2^	5.8 × 10 ^−3^	5.9 × 10 ^−4^		3.8 × 10 ^−4^	1.7 × 10 ^−4^	2.3 × 10 ^−4^
32	6.2 × 10 ^−4^	5.6 × 10 ^−4^	1.1 × 10 ^−2^		2.4 × 10 ^−4^	2.0 × 10 ^−4^	1.8 × 10 ^−4^
64	0.00047	1.6 × 10 ^−3^	2.1 × 10 ^−1^		2.7 × 10 ^−4^	1.7 × 10 ^−4^	0.00013

**Table 2 sensors-23-06778-t002:** MSE errors for LSTM models for the square set-point signal. The blue background indicates the best-performing models of each type with the lowest error value.

	${\mathrm{LSTM}}_{\mathrm{sq}}$				${\mathrm{LSTM}}_{\mathrm{sq}}^{\mathrm{SP}}$		
$n_{N}$	$n_{A}=1$	$n_{A}=2$	$n_{A}=3$		$n_{A}=1$	$n_{A}=2$	$n_{A}=3$
8	2.5 × 10 ^−1^	2.8 × 10 ^−1^	1.6 × 10 ^−1^		1.0 × 10 ^−3^	9.4 × 10 ^−4^	0.00053
12	3.3 × 10 ^−1^	2.3 × 10 ^−1^	2.2 × 10 ^−1^		5.9 × 10 ^−4^	8.2 × 10 ^−4^	9.5 × 10 ^−4^
16	3.4 × 10 ^−1^	1.4 × 10 ^−1^	3.0 × 10 ^−1^		6.5 × 10 ^−4^	6.9 × 10 ^−4^	6.4 × 10 ^−4^
24	6.7 × 10 ^−2^	6.2 × 10 ^−2^	2.5 × 10 ^−2^		6.3 × 10 ^−4^	7.6 × 10 ^−4^	7.3 × 10 ^−4^
32	1.1 × 10 ^−1^	5.1 × 10 ^−2^	7.8 × 10 ^−2^		6.3 × 10 ^−4^	6.6 × 10 ^−4^	7.2 × 10 ^−4^
64	7.8 × 10 ^−2^	6.3 × 10 ^−2^	0.023		6.9 × 10 ^−4^	6.2 × 10 ^−4^	6.0 × 10 ^−4^

## Data Availability

On request from the authors.

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
