# Peer review of "Forgery Cyber-Attack Supported by LSTM Neural Network: An Experimental Case Study"

_sensors, 2023, doi:10.3390/s23156778_

Round 1

Reviewer 1 Report

The topic of the paper is interesting and novel, although the state of the art is very short and general. The paper should have better description of problem definition, how does it solve by other teams, formulate scientific questions to be solved and define the methodology. Also, the discussion of results can be then discussed a compared it with promised outputs and other methods. So the paper should be mainly improved and extended. Authors can consider to focus more on specific method and discussed them in more details, rather then write just general description.

Minor editing of English language required

Reviewer 2 Report

How does the use of Long Short-Term Memory (LSTM) neural networks in cyber-attacks exploit the vulnerability of industrial control systems?

What are some potential protection methods suggested in the research to mitigate the risks of forgery cyber-attacks supported by LSTM neural networks?

How do cyber-attacks on sensor communication pose a significant threat to IoT and Industrial IoT devices? Provide specific examples of such attacks.

Explain the role of sensors in cyber-security and control algorithms, particularly in detecting malicious activity and anomalies.

What are the advantages of LSTM networks over classical recurrent networks in modeling dynamical processes?

How does the research address the issue of human error in the context of cyber-attacks on industrial networks?

Describe the experimental setup and process used to train and evaluate the LSTM neural network in the laboratory control network.

What are the potential implications of carrying out prolonged cyber-attacks that are difficult to detect by operators on industrial processes?

Discuss the limitations and challenges associated with identifying signals relevant to the operation of the process and developing accurate models for detecting cyber-attacks based on network traffic.

How do the findings of this research contribute to the understanding of the vulnerability of industrial control systems and the need for improved security measures?

Need some minor Language Editing

Reviewer 3 Report

This paper examines how vulnerable a network industrial control system is to cyber-attacks, with a focus on the MAGLEV process. The authors use LSTM neural networks to simulate the process and study how a cyber-attack can disrupt its operation. They also suggest possible protection methods against such attacks. The paper provides a detailed account of their methodology, model training, evaluation, and selection, and discusses the overall data flow of the network. The authors aim to shed light on the vulnerability of industrial control systems to cyber-attacks and propose ways to protect against them. Overall, the paper is well-written and structured, but there are some typos and grammatical errors that need to be corrected. Additionally, the authors should update the state of the art with similar works in the area (see [1],[2],[3]). In section 4.2, it is unclear which types of cyber-attacks the proposed methodology protects against. Finally, the authors should explain why their proposed method achieves better results compared to existing methodologies for revealing temporal uncorrelated attacks.([4])

[1]Wilson, Mitchell & Mahmood, Hisham & Giordano, Joseph. (2021). Detection and Mitigation of Cyberattacks Against Power Measurement Channels Using LSTM Neural Networks. 1419-1426.

[2]Gao, Jun & Gan, Luyun & Buschendorf, Fabiola & Zhang, Liao & Liu, Hua & Li, Peixue & Dong, Xiaodai & Lu, Tao. (2019). LSTM for SCADA Intrusion Detection. 10.13140/RG.2.2.30054.16962.

[3]Kotenko, Igor & Lauta, Oleg & Kribel, Kseniya & Saenko, Igor. (2021). LSTM Neural Networks for Detecting Anomalies Caused by Web Application Cyber Attacks. 10.3233/FAIA210014.

4. J. Gao et al., "LSTM for SCADA Intrusion Detection," 2019 IEEE Pacific Rim Conference on Communications, Computers and Signal Processing (PACRIM), Victoria, BC, Canada, 2019, pp. 1-5, doi: 10.1109/PACRIM47961.2019.8985116.

Overall, the paper is well-written and structured, but there are some typos and grammatical errors that need to be corrected.

Round 2

Reviewer 1 Report

The revision of the paper made the paper more clear, also the state of the art was improved. Last but not least, there is a better discussion of results and presented contribution.

Minor editing of English language required.

Author Response

Thank you for your thorough inspection of this paper. We are convinced that the paper is greatly improved thanks to the in-depth reviews. As suggested, we have rechecked the text for editing and language errors, and we have fixed some minor mistakes.